# Comparative efficacy of opioid and non-opioid analgesics in labor pain management: A network meta-analysis

Yiru Chen[1,2], Hongchun Chen[1,2,3], Chunhui Yuan[1,2]*

1 Department of Clinical Medicine, School of Medicine, Hangzhou City University, Hangzhou, Zhejiang, China, 2 Key Laboratory of Novel Targets and Drug Study for Neural Repair of Zhejiang Province, School of Medicine, Hangzhou City University, Hangzhou, Zhejiang, China, 3 College of Pharmaceutical Science, Zhejiang University of Technology, Hangzhou, Zhejiang, China

* ch_yuan@zju.edu.cn

## Abstract

### Background

Effective labor pain management is crucial for parturient well-being, as it can improve the delivery experience of pregnant women and reduce anxiety and tension. This systematic review and network meta-analysis compared the efficacy and safety of various analgesics, classified by drug category and individual treatment methods, for labor pain control.

### Methods

A comprehensive literature search was conducted in Pubmed, EMBASE, Cochrane Library, and Web of Science databases. All searches commenced from the database's inception to the date of the literature search (May 31, 2023). The Cochrane Risk of Bias 2 tool assessed study bias risk. Network meta-analyses using a random-effects model and odds ratios (ORs) with 95% confidence intervals (CIs) were performed.

### Results

Fifteen randomized controlled trials evaluating analgesic interventions in ASA I or II parturients were included. Combination therapies (OR: 5.81; 95% CI, 3.76–7.84; probability: 60%) and non-opioid analgesics (OR: 5.61; 95% CI, 2.91–8.30; probability: 39.2%) were superior to placebo for labor pain relief. Specifically, dexmedetomidine/ropivacaine/sufentanil (OR: 7.32; 95% CI, 2.73–11.89; probability: 40.6%) and dexmedetomidine/ropivacaine (OR: 6.50; 95% CI, 2.51–10.33; probability: 11.9%) combinations, bupivacaine/fentanyl and ropivacaine/sufentanil combinations, and remifentanil monotherapy showed improved analgesic efficacy versus placebo. Dexmedetomidine/ropivacaine reduced parturient nausea and vomiting versus alternatives.

**Data Availability Statement:** All relevant data are within the paper and its Supporting information files.

**Funding:** The author(s) received no specific funding for this work.

**Competing interests:** The authors have declared that no competing interests exist.

## Conclusion

Non-opioids, opioids and combinations thereof effectively relieved labor pain. In addition, dexmedetomidine/ropivacaine combination demonstrated analgesic efficacy and lower nausea and vomiting incidence.

## Introduction

In the field of obstetrics, pain management for women during labor has consistently been recognized as a significant and challenging issue [1, 2]. During labor, intense pain can be induced by uterine contractions and the anxiety and fear experienced by the woman [1]. With the advancement of medical technology, there has been an emergence of various medications and multiple drug delivery methods. During labor, local anesthetics and controlled release analgesics are administered axially across the nerve (intrathecal or epidural) to reduce pain, which enables achieving the desired analgesic effect while minimizing the dosage of each drug, thereby reducing the risk of adverse reactions [3, 4]. The utilization of low concentrations of local anesthetics can decrease motor nerve blockage, whereas the use of low concentrations of opioid analgesics can minimize the overall impact on both the mother and the fetus [5].

In addition, intravenous administration of opioid analgesics has been used in some clinical trials. However, it has shown a limited impact on maternal pain scores and its analgesic effects are not consistently reliable. Moreover, it is frequently accompanied by adverse reactions such as nausea and vomiting [6]. However, remifentanil, as an ultra-short-acting opioid, can deliver satisfactory analgesic effects only when used for patient-controlled intravenous analgesia during labor [7]. Clearly, the analgesic effect of a given medication may vary depending on the particular pain management approach utilized for women in labor.

The selection of appropriate medication or combination therapy is crucial for the management of women's pain in labor, yet it remains an unresolved clinical challenge. Although several studies have provided preliminary evidence of the effectiveness and safety of opioid analgesics in labor analgesia [8–10]. However, there is still insufficient systematic review and evidence to compare the impact of different treatment strategies on women's labor pain management.

The American Society of Anesthesiologists (ASA) divides patients into six classifications based on the severity of coexisting diseases and functional status before surgery, which plays a role as a risk predictor. ASA Physical Status classification I or II are defined as A normal healthy patient and A patient with mild systemic disease. Although pregnancy is not a disease, the parturient's physiologic state is significantly altered from when the woman is not pregnant, hence the assignment of ASA Physical Status II for a woman with uncomplicated pregnancy. Obstetric examples, including, but Not limited to the Following: Normal pregnancy, well-controlled gestational hypertension, controlled pre-eclampsia without severe features, diet-controlled gestational diabetes mellitus [11].

This study utilizes a network meta-analysis to integrate all available direct and indirect evidence from multiple clinical trials, comprehensively comparing the effects of various treatment methods on labor pain in women with ASA classification I or II. The objective of this network meta-analysis is to systematically evaluate the effectiveness of different treatment strategies in controlling labor pain in women with ASA classification I or II, providing the scientific basis and evidence-based support for obstetricians in selecting the optimal labor pain management approach. The main outcomes of interest is pain scores. The secondary outcome of interest is drug side effects.

## Methods

This network meta-analysis adheres to the guidelines provided by Preferred Reporting Items for Systematic Reviews and Meta-Analyses (PRISMA) [12]. It is based on aggregated data, and the review protocol has been registered with PROSPERO(CRD42023417670). Because there were no investigations involving human subjects or use of patient data for research purposes, approval under Declaration of the World Medical Association was not required (S3 Table in S3 File).

### Literature search

We conducted a systematic search of four databases, namely Pubmed, EMBASE, Cochrane Library, and Web of Science, covering literature from the inception of each database until May 2023. The search strategy included relevant keywords and their synonyms, such as "obstetric analgesia," "labor pain," "epidural analgesia," "opioid drugs," "local anesthetics," "remifentanil," "sufentanil," "midazolam," etc. No language restrictions were applied to include studies in multiple languages (S1 File).

In addition to the database search, we manually screened the reference lists of relevant systematic reviews and meta-analyses for further potential studies. Furthermore, we searched the ClinicalTrials.gov website and conducted a Google Scholar search to identify ongoing or completed randomized controlled trials. To ensure that no same study, we used EndNote software to remove duplicate records obtained from all sources.

No language restrictions were applied to avoid missing any relevant studies. In necessary cases, we will attempt to use translation software or seek assistance from language experts to translate and screen non-English literature.

### Study selection

Two authors independently screened the relevant records to determine their eligibility for inclusion based on the predetermined criteria. In cases of disagreement, the original articles were reviewed again, and discussions were held to reach a consensus.

**Inclusion criteria.**

1. Study population: Pregnant women with a normal pregnancy and undergoing imminent delivery, with an American Society of Anesthesiologists (ASA) classification of I or II.

2. Study design: Only randomized controlled trials (RCTs) will be included.

3. Intervention: Combination medications, monotherapies, opioids and non-opioids medications.

4. Outcome measures: The main outcomes of interest is pain scores. The secondary outcome of interest is drug side effects.

**Exclusion criteria.**

1. Non-randomized controlled trials, case reports, case series, and non-human studies.

2. Conference abstracts, reviews, or meta-analyses.

3. Studies without extractable data (including mean values and standard deviations of pain scores).

4. Literature lacking details of the study methods, intervention measures, or outcome data.

5. Duplicate publications. For multiple related studies published by the same author, we will assess if they are different reports of the same study and, if necessary, contact the authors for clarification. The most comprehensive report will be selected for analysis.

6. Studies with significant limitations or biases in their research process or results that pose a substantial threat to the internal and external validity of the study. This will be assessed through a comprehensive evaluation while reading the full-text articles.

### Risk-of-bias assessment

Risk of bias in the included trials was assessed using the Cochrane Risk of Bias Version 2 (ROB 2) tool [13]. This tool evaluates five bias domains: random sequence generation, allocation concealment, participant/personnel blinding, incomplete outcome data, and selective outcome reporting. Each domain was judged as low risk, some concerns or high risk based on pre-defined criteria.

Two independent reviewers evaluated risk of bias in all included trials, with disagreements resolved via discussion and consensus. Judgments were made for each trial based on the ROB 2 guidelines, while considering study characteristics and clinical expertise. Trials rated as "high risk" in one or more key domains were deemed at overall high risk of bias; while trials rated as "low risk" in most/all domains were deemed at overall low risk of bias. Judgments are based on, and summarise, the answers to signalling questions.

### Statistical analysis

Statistical analyses were performed using R 4.2.2 with the package "gemtc" and "netmeta" and STATA 15.1. Study characteristics including design, interventions, sample size and outcome measures were extracted and summarized. Quantitative heterogeneity across trials was assessed using the $I^2$ statistic. In case of low heterogeneity ($I^2 < 50\%$), a fixed-effects model was utilized for network meta-analysis. Otherwise, a random-effects model was employed for sensitivity analysis and subgroup analysis to explore potential sources of heterogeneity. Effect sizes such as odds ratios (ORs), mean differences (MDs) and their 95% confidence intervals (CIs) were calculated to evaluate treatment efficacy.

Pain scores were collected at specified timepoints e.g. 20, 30 and 60 minutes after analgesia initiation or at the second stage of labor onset. Scores from better analgesic effect groups were included in analysis. When available, average pain scores throughout the entire labor process were preferentially analyzed. Pain rating scales and scores were transformed or standardized for facile comparison.

Potential publication bias was evaluated via funnel plot analysis. Sensitivity analyses omitting one study at a time were conducted to determine individual study influence on overall results. Where necessary, subgroup analyses were performed to establish the effect of study characteristics and other factors.

## Results

### Study selection

We obtained a total of 633 relevant studies through our literature search. After excluding 148 duplicate publications, we proceeded to title and abstract screening of 485 studies. Finally, we included 15 randomized controlled trials for quantitative analysis, among which 10 studies were grouped based on the type of medication [14–28]. Six studies reported the use of double-

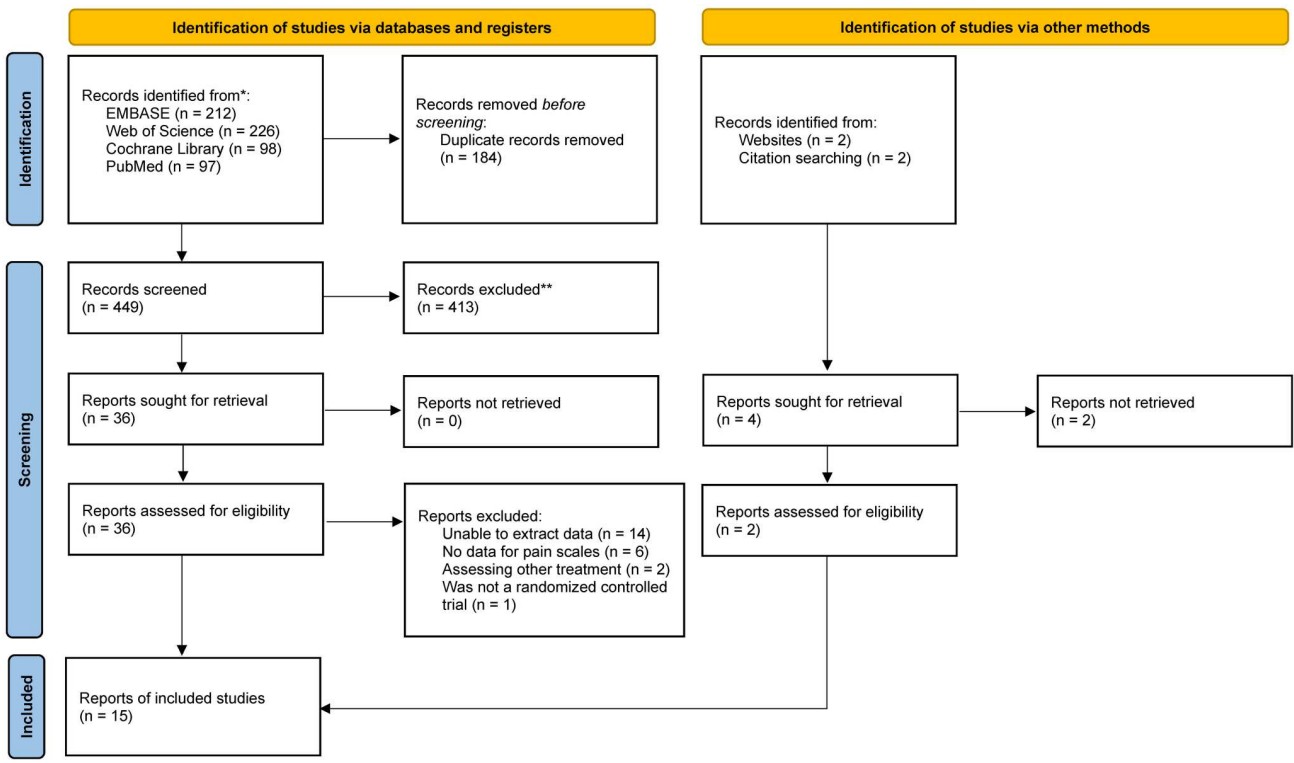

**Fig 1. PRISMA flow diagram of study inclusion and exclusion.**

blinding, while five studies did not mention blinding. The included studies originated from 7 countries or regions. The total number of participants in the included studies was 2466. Please refer to Fig 1 for the specific flowchart of the literature screening process.

The 15 studies included in our network meta-analysis were organized by drug categories [14–28]. All those studies reported pain scale scores during childbirth. There were 12 two-group studies and 3 three-group studies in total. The average age of the participants was 28.14 years, with an average height of 162.11 cm and an average weight of 80.17 kg.

The intervention measures in the included studies mainly included four categories: placebo (3 randomized controlled trials), non-opioid medications (3 randomized controlled trials), opioid medications (8 randomized controlled trials), and combination medications (10 randomized controlled trials). The treatment typically started when the cervical dilation reached around 3 centimeters.

The primary outcome measure was the pain relief score calculated from the pain ratings (using the visual analog scale, VAS [14], or numeric rating scale, NRS [1]) at 20, 30, or 60 minutes after the onset of the first stage of labor or at the start of the second stage of labor. The secondary outcome of interest is drug side effects, such as nausea and vomiting. The main characteristics and interventions of the included studies are summarized in Table 1.

## Risk-of-bias and quality-of-evidence assessment

According to the ROB2 tool assessment, seven out of the total studies (47%) were found to have some degree of bias risk, the remaining studies had a low risk of bias. When analyzing the

**Table 1. The characteristics of included studies.**

| Study | Country | Scale | Group | No. of participant | Treatment | Outcomes |
|---|---|---|---|---|---|---|
| Tveit(15) | Norway | VAS | RA | 17 | Remifentanil dose: 0.15 μg kg$^{-1}$, increased by 0.15 μg kg$^{-1}$. | Analgesic effect |
| | | | EA | 20 | PCA pump for 2 min lock time, 2ml min$^{-1}$ injection rate. | First stage of labor |
| 2012 | | | | | Continuous epidural infusion of ropivacaine 1 mg ml$^{-1}$ | Second stage of labor |
| | | | | | and fentanyl 2 μg ml$^{-1}$. Initial bolus: 10 ml, 5 ml top-up | Nausea |
| | | | | | after 5 min. Infusion rate: 10 ml h$^{-1}$. | |
| Evron(16) | Israel | VAS | R | 43 | Remifentanil via PCIA: 20 μg starting dose, 3 min | Analgesic effect |
| | | | M | 45 | lockout. 5 μg increments on request. | First stage of labor |
| 2005 | | | | | Meperidine: 75 mg in 100 mL saline over 30 min | Second stage of labor |
| | | | | | (approx. 1 mg kg$^{-1}$ single bolus). | Nausea |
| Douma(17) | Netherlands | VAS | R | 60 | Remifentanil: 40 μg loading dose, 40 μg bolus$^{-1}$, 2 min | Analgesic effect |
| | | | P | 60 | lockout. | First stage of labor |
| 2010 | | | F | 60 | Perhidine: 49.5 mg loading dose, 5 mg bolus$^{-1}$, 10 min | Second stage of labor |
| | | | | | lockout. | Nausea |
| | | | | | Fentanyl: 50 μg loading dose, 20 μg bolus$^{-1}$, 5 min | |
| | | | | | lockout, max 240 μg h$^{-1}$. | |
| Stocki(18) | Jerusalem | NRS | PCIA | 19 | Remifentanil bolus: 20–60 μg. PCIA lockout: 2 min, | Analgesic effect |
| | | | EA | 20 | adjustable to 1 min. | First stage of labor |
| 2014 | | | | | Initial: 15 ml 0.1% bupivacaine + 50 μg fentanyl. PCEA: | Second stage of labor |
| | | | | | 0.1% bupivacaine + 2 μg ml$^{-1}$ fentanyl, 5 ml h$^{-1}$ basal, | Nausea |
| | | | | | 10 ml bolus, 20 min lockout. | |
| Douma(19) | Netherlands | VAS | R | 10 | Remifentanil 40 μg loading dose, 40 μg bolus$^{-1}$, 2 min | Analgesic effect |
| | | | EA | 10 | lockout, 36 sec bolus duration via Graseby 3300 pump. | First stage of labor |
| 2011 | | | | | Loading dose: 0.2% ropivacaine 12.5 mL. Continuous | Second stage of labor |
| | | | | | infusion: 0.1% ropivacaine + sufentanil 0.5 μg ml$^{-1}$ at | Nausea |
| | | | | | 10 ml h$^{-1}$. | |
| Ismail(20) | Egypt | VAS | EA/CSE | 760 | Levobupivacaine + fentanyl via epidural catheter or | Analgesic effect |
| | | | PCIA | 380 | injected intrathecally. Continuous infusion: 8 ml h$^{-1}$ via | First stage of labor |
| 2012 | | | | | electronic pump. | Second stage of labor |
| | | | | | PCIA: 0.1 μg kg$^{-1}$ remifentanil bolus in 1 min. Lockout: | Nausea |
| | | | | | 1 min. Subsequent dosage increases. | |
| Jia(21) | China | VAS | A | 40 | IV analgesia pump: remifentanil initial/background/PCA | Analgesic effect |
| | | | B | 40 | dose: 0.25/0.05/0.25 μg kg$^{-1}$. Locking time: 2 min. | First stage of labor |
| 2020 | | | C | 40 | 1.5 mg ropivacaine + 2.5 μg sufentanil injected into | Second stage of labor |
| | | | | | interspinal interstice. Electronic pump: 6 ml h$^{-1}$ | Nausea |
| | | | | | background, 2 ml rapid dose, 15 min lockout. | |
| | | | | | Spontaneous labor and routine obstetric treatment. | |
| Rezk(22) | Egypt | VAS | F | 40 | 50 μg fentanyl in 18 ml saline (20 ml total) over 10 min | Analgesic effect |
| | | | P | 40 | IV infusion. | Nausea |
| 2015 | | | | | 100 mg pethidine by intramuscular injection. | |
| Nunes(23) | Brazil | VAS | D | 100 | IV 0.25 mg kg$^{-1}$ pethidine. | Analgesic effect |
| | | | P | 100 | IV 25 mg kg$^{-1}$ dipyrone. | Nausea |
| 2019 | | | | | | |
| Zhao(24) | China | VAS | R | 40 | Epidural 0.125% ropivacaine. | Analgesic effect |
| | | | D | 40 | Epidural 0.125% ropivacaine with dexmedetomidine | First stage of labor |
| 2017 | | | | | (0.5 μg kg$^{-1}$ as bolus only). | Second stage of labor |
| | | | | | | Nausea |

*(Continued)*

**Table 1.** (Continued)

| Study | Country | Scale | Group | No. of participant | Treatment | Outcomes |
|---|---|---|---|---|---|---|
| Wang(25) | China | VAS | R | 75 | An epidural pump used 0.1% ropivacaine. First dose: 10 | Analgesic effect |
| | | | R+Y | 75 | ml. Infusion: 10 ml h$^{-1}$. Additional: 5 ml. | First stage of labor |
| 2018 | | | | | Same as group R, but with a mixture of 0.1% | Second stage of labor |
| | | | | | ropivacaine and 0.5 µg ml$^{-1}$ dexmedetomidine. | Nausea |
| Karadjova(26) | North Macedonia | VAS | RG | 80 | PCA with 2-minute lock. Remifentanil started small, | Analgesic effect |
| | | | EG | 75 | increased from 0.2 to 1 µg kg$^{-1}$. | Nausea |
| 2019 | | | | | Bolus: 10 ml 0.1% Bupivacain + 0.05 mg Fentanil. | |
| | | | | | Epidural bolus: 10 ml 0.0625% Bupivacain + 2 µg ml$^{-1}$ | |
| | | | | | Fentanil every 60 min. | |
| Li(27) | China | VAS | RS | 35 | Epidural: 10 ml 0.5 µg ml$^{-1}$ sufentanil + 0.1% ropivacaine. | Analgesic effect |
| | | | RD | 36 | Maintenance: Apon PCA pump at 7 ml h$^{-1}$. | First stage of labor |
| 2020 | | | RDS | 36 | Epidural: 10 ml of 0.5 µg ml$^{-1}$ Dex + 0.1% ropivacaine. | Second stage of labor |
| | | | | | Maintenance: Apon PCA pump at 7 ml h$^{-1}$. | Nausea |
| | | | | | Epidural: 10 ml of 0.25 µg ml$^{-1}$ Dex + 0.25 µg ml$^{-1}$ | |
| | | | | | sufentanil + 0.1% ropivacaine. Maintenance: Apon PCA | |
| | | | | | pump at 7 ml h$^{-1}$. | |
| Cheng(28) | China | VAS | RD1 | 80 | EA: 100 ml ropivacaine + 0.5 µg ml$^{-1}$ dexmedetomidine. | Analgesic effect |
| | | | RS1 | 80 | Pump: 10 ml load, 8 ml h$^{-1}$ infusion, 8 ml PCA with 30 | First stage of labor |
| 2019 | | | | | min lockout. | Second stage of labor |
| | | | | | EA: 100 mL ropivacaine + 0.5 µg ml$^{-1}$ sufentanil. Pump: | Nausea |
| | | | | | 10 ml load, 8 ml h$^{-1}$ infusion, 8 ml PCA with 30 min | |
| | | | | | lockout. | |
| Wu(29) | China | VAS | CSEA | 90 | Subarachnoid: 5 µg sufentanil. Total: 120 ml. Infusion: | Analgesic effect |
| | | | C | 90 | 6 ml h$^{-1}$ ropivacaine 75 mg + sufentanil 45 µg. Single: 8 | Second stage of labor |
| 2022 | | | | | ml. Lock: 15 min. | |
| | | | | | Spontaneous labor and routine obstetric treatment. | |

PCA, Patient-Controlled Analgesia; PCIA, Patient-Controlled Intravenous Analgesia; PCEA, Patient-Controlled Epidural Analgesia; IV, Intravenous Injection; EA, Epidural Anesthesia; VAS, Visual Analogue Scale; NRS, Numerical Rating Scal

bias risk based on individual domains, all categories demonstrated low bias risk in over 50% of the included studies (S2 Fig in S2 File).

## Pain score

**Analgesic medications classes.** A total of 10 studies were included in the Analgesic medications network meta-analysis [14, 17–20, 22, 25–28]. Analgesic drug therapy is divided into four categories: non-opioid analgesics (Class A; 3 RCTs), opioid analgesics (Class B; 8 RCTs), combination therapy (Class C; 10 RCTs), and placebo (Class D; 3 RCTs) (Fig 2A). In the network meta-analysis, the most interactions were between combination therapy and opioid analgesics (6 interactions), followed by combination therapy and non-opioid analgesics (2 interactions), combination therapy and placebo (2 interactions), opioid drugs and non-opioid analgesics (1 interaction), and opioid analgesics and placebo (1 interaction)(Fig 2B). Heterogeneity analysis revealed significant heterogeneity at $I^2$ = 97.91% (Fig 2C), therefore, a random-effects model was employed for network meta-analysis.

The results showed that non-opioid analgesics (Class A vs D; OR, 5.61; 95% CI, 2.91–8.30), opioid analgesics (Class B vs D; OR, 4.46; 95% CI, 2.22–6.59), and combination therapy (Class

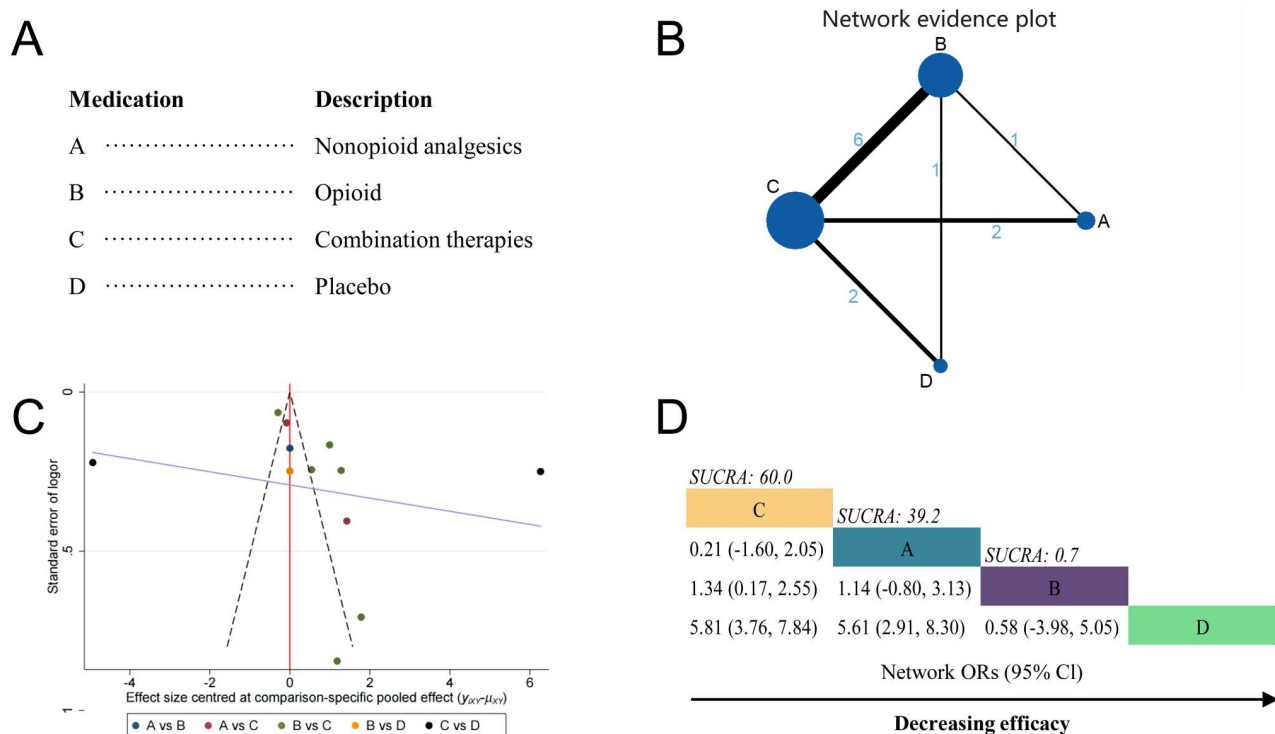

**Fig 2. Network Meta-analysis by drug category.** (A) Description of drug categories included in the network meta-analysis. (B)Network graph showing overall treatment effect comparisons between nodes (blue circles), where each node represents a drug category or placebo. The size of each node is proportional to the total number of participants randomized to receive the drug category. The width of each connecting line is proportional to the number of trial-level comparisons between the two nodes. (C) Funnel plot of publication bias, comparing publication bias between drug categories. (D) Schematic diagram listing the most effective drug categories globally according to surface under the cumulative ranking curve (SUCRA) analysis. CI, confidence interval; OR, odds ratio.

C vs D; OR, 5.81; 95% CI, 3.76–7.84) were all significantly superior to placebo in terms of pain relief. However, in pairwise comparisons, no one drug category was favored over another. Surface under the cumulative ranking curve analysis (SUCRA) analysis showed that combination therapy had the highest cumulative ranking for relieving labor pain (Class C; SUCRA = 60%), followed by non-opioid analgesics (Class A; SUCRA = 39.2%) and opioid analgesics (Class B; SUCRA = 0.7%; Fig 2D).

**Analgesic medications.** A total of 15 studies were included in this network meta-analysis [14–22, 24–29]. The 11 treatments were referred to as: dipyrone (Class A; 1 RCT), fentanyl (Class B; 3 RCTs), pethidine (Class C; 5 RCTs), remifentanil (Class D;10 RCTs), ropivacaine (Class E; 2 RCTs), placebo (Class F; 3 RCTs), bupivacain+fentanyl (Class G; 3 RCTs), ropivacaine+sufentanil (Class H; 7 RCTs), ropivacaine+fentanyl (Class I; 1 RCT), dexmedetomidine +ropivacaine (Class J; 5 RCTs) and dexmedetomidine+ropivacaine +sufentanil (Class K; 2 RCTs) (Fig 3A).

In the network meta-analysis, the most interactions were between bupivacain+fentanyl and remifentanil (3 interactions), with the rest having either 1 or 2 interactions (Fig 3B). We used a random-effects model because the heterogeneity analysis revealed significant heterogeneity as $I^2$ = 97.92% (Fig 3C). The results showed that compared to placebo, the combination of bupivacaine+fentanyl (Class G vs F; OR, 5.90; 95% CI, 1.81–9.84), dexmedetomidine+ropivacaine (Class J vs F; OR, 6.50; 95% CI, 2.51–10.33), dexmedetomidine+ropivacaine+sufentanil (Class

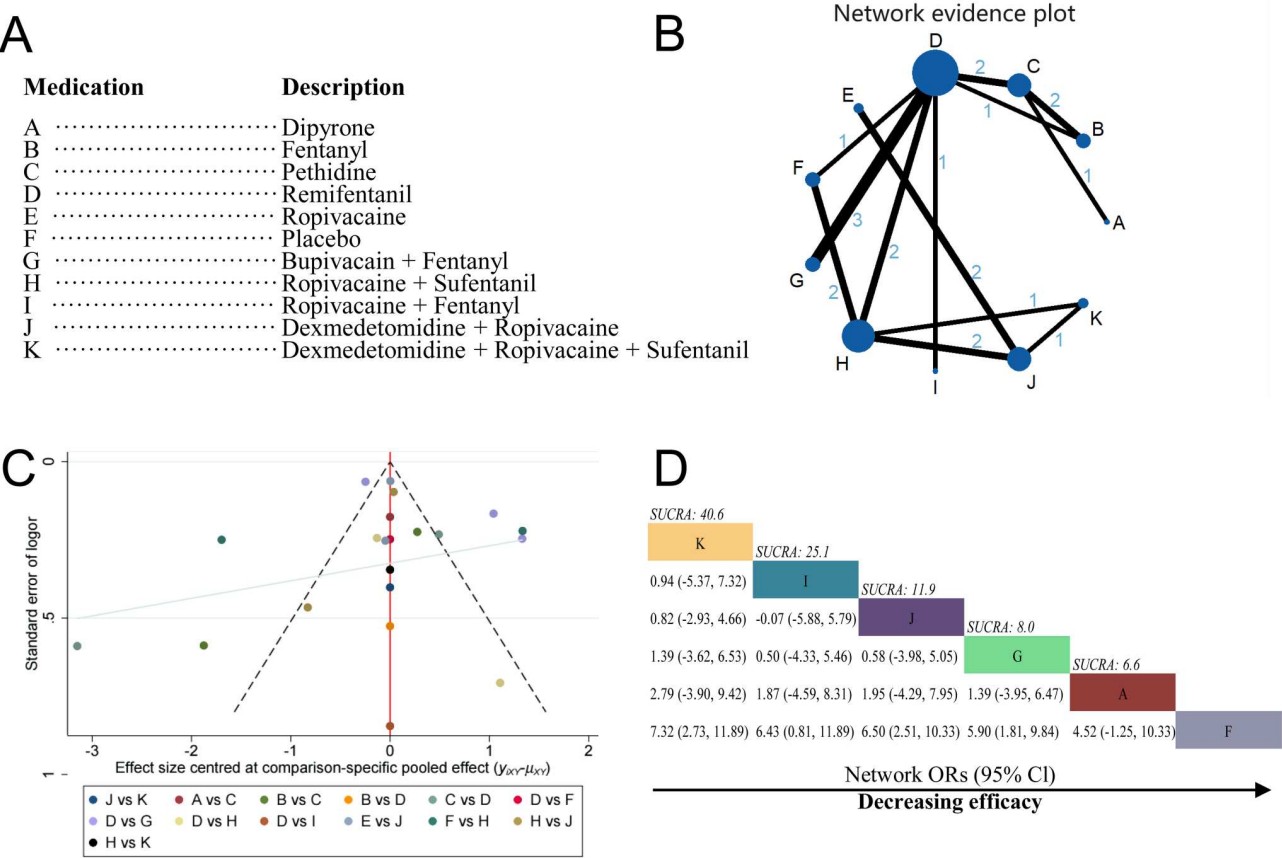

**Fig 3. Network meta-analysis by individual treatment.** (A) Description of drug categories included in the network meta-analysis. (B) Network graph showing overall treatment effect comparisons between nodes (blue circles), where each node represents a drug category or placebo. The size of each node is proportional to the total number of participants randomized to receive the drug category. The width of each connecting line is proportional to the number of trial-level comparisons between the two nodes. (C) Funnel plot of publication bias, comparing publication bias between drug categories. (D) Schematic diagram listing the most effective drug categories globally according to surface under the cumulative ranking curve (SUCRA) analysis. CI, confidence interval; OR, odds ratio.

K vs F; OR, 7.32; 95% CI, 2.73–11.89), ropivacaine+sufentanil (Class H vs F; OR, 5.86; 95% CI, 3.13–8.54), and remifentanil monotherapy (Class B vs F; OR, 4.40; 95% CI, 1.08–7.64) significantly improved pain in parturient women. Comparisons of fentanyl+ropivacaine combination, meperidine, fentanyl, pethidine, and ropivacaine with placebo did not demonstrate significant advantages in pain control. There were no significant differences observed between the remaining groups. SUCRA analysis showed that dexmedetomidine+ropivacaine+sufentanil had the highest cumulative ranking for relieving labor pain (Class K; SUCRA = 40.6%), followed by ropivacaine+fentanyl (Class I; SUCRA = 25.1%), dexmedetomidine+ropivacaine (Class J; SUCRA = 11.9%), bupivacaine+fentanyl (Class G; SUCRA = 8.0%), and dipyrone (Class A; SUCRA = 6.6%)(Fig 3D).

## Safety

**Analgesic medications classes.** A total of 9 studies were included in the analgesic medications classes network meta-analysis [14, 17–20, 22, 25–27]. Four distinct drug categories were encompassed, as depicted in Fig 4A. These were non-opioid drugs (Class A; 3 RCTs), opioid

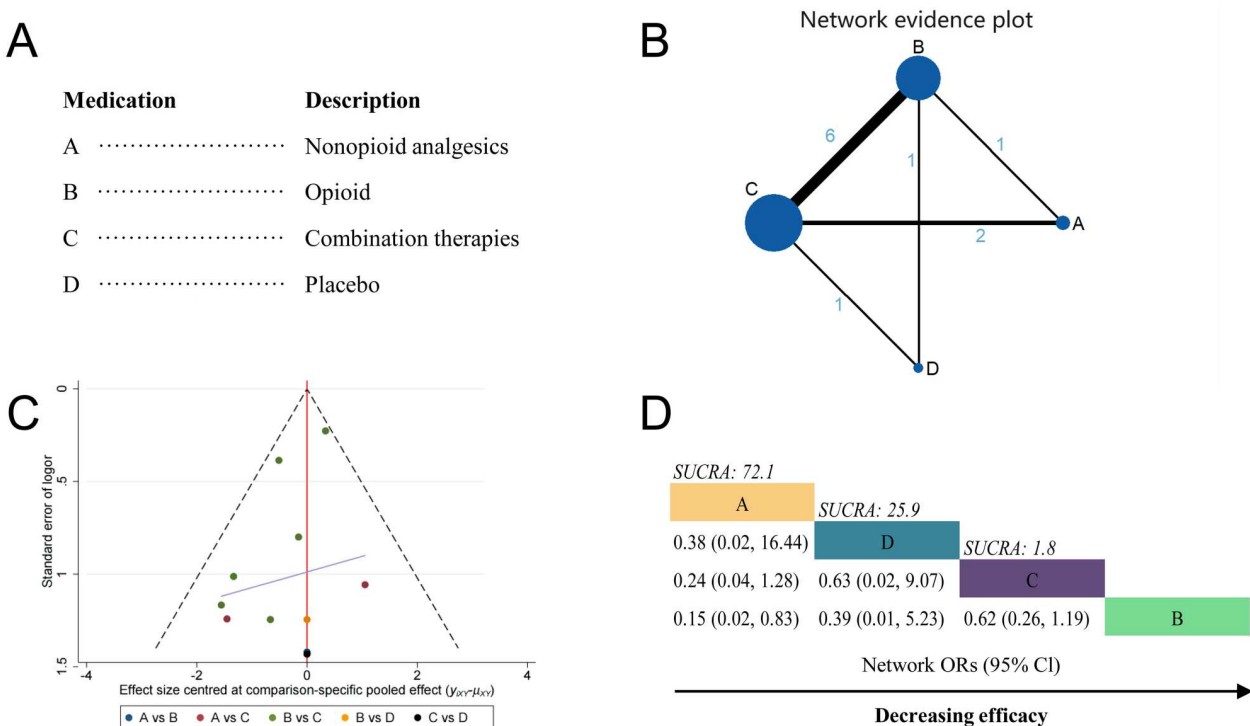

**Fig 4. Network meta-analysis by drug category.** (A) Description of drug categories included in the network meta-analysis. (B)Network graph showing overall treatment effect comparisons between nodes (blue circles), where each node represents a drug category or placebo. The size of each node is proportional to the total number of participants randomized to receive the drug category. The width of each connecting line is proportional to the number of trial-level comparisons between the two nodes. (C) Funnel plot of publication bias, comparing publication bias between drug categories. (D) Schematic diagram listing the most effective drug categories globally according to surface under the cumulative ranking curve (SUCRA) analysis. CI, confidence interval; OR, odds ratio.

drugs (Class B; 8 RCTs), combination therapy (Class C; 9 RCTs), and placebo (Class D; 2 RCTs).

As the result of the network meta-analysis, the most interactions were between combination therapy and opioid drugs (6 interactions), followed by combination therapy and non-opioid drugs (2 interactions), combination therapy and placebo (1 interaction), opioid drugs and non-opioid drugs (1 interaction), and opioid drugs and placebo (1 interaction; Fig 4B). The $I^2$ was 36.4%, indicating moderate heterogeneity (Fig 4C). Here, we used a random-effects model for network meta-analysis. The results showed no differences in the occurrence rate of maternal nausea and vomiting reactions among the different drug categories. SUCRA analysis showed that non-opioid analgesics had the highest cumulative ranking for reducing the side effect of nausea and vomiting in parturients (Class A; SUCRA = 72.1%), followed by placebo (Class D; SUCRA = 25.9%) and combination therapy (Class C; SUCRA = 1.8%; Fig 4D).

**Analgesic medications.** A total of 14 studies were included in this network meta-analysis [14–22, 24–27, 29] which covered 11 distinct drug categories: dipyrone (Class A; 1 RCT), fentanyl (Class B; 3 RCTs), pethidine (Class C; 5 RCTs), remifentanil (Class D;10 RCTs), ropivacaine (Class E; 2 RCTs), placebo (Class F; 2 RCTs), bupivacain+fentanyl (Class G; 2 RCTs), ropivacaine+sufentanil (Class H; 7 RCTs), ropivacaine+fentanyl (Class I; 1 RCT), dexmedetomidine+ropivacaine (Class J; 5 RCTs) and dexmedetomidine+ropivacaine+sufentanil (Class K; 2 RCTs; Fig 5A).

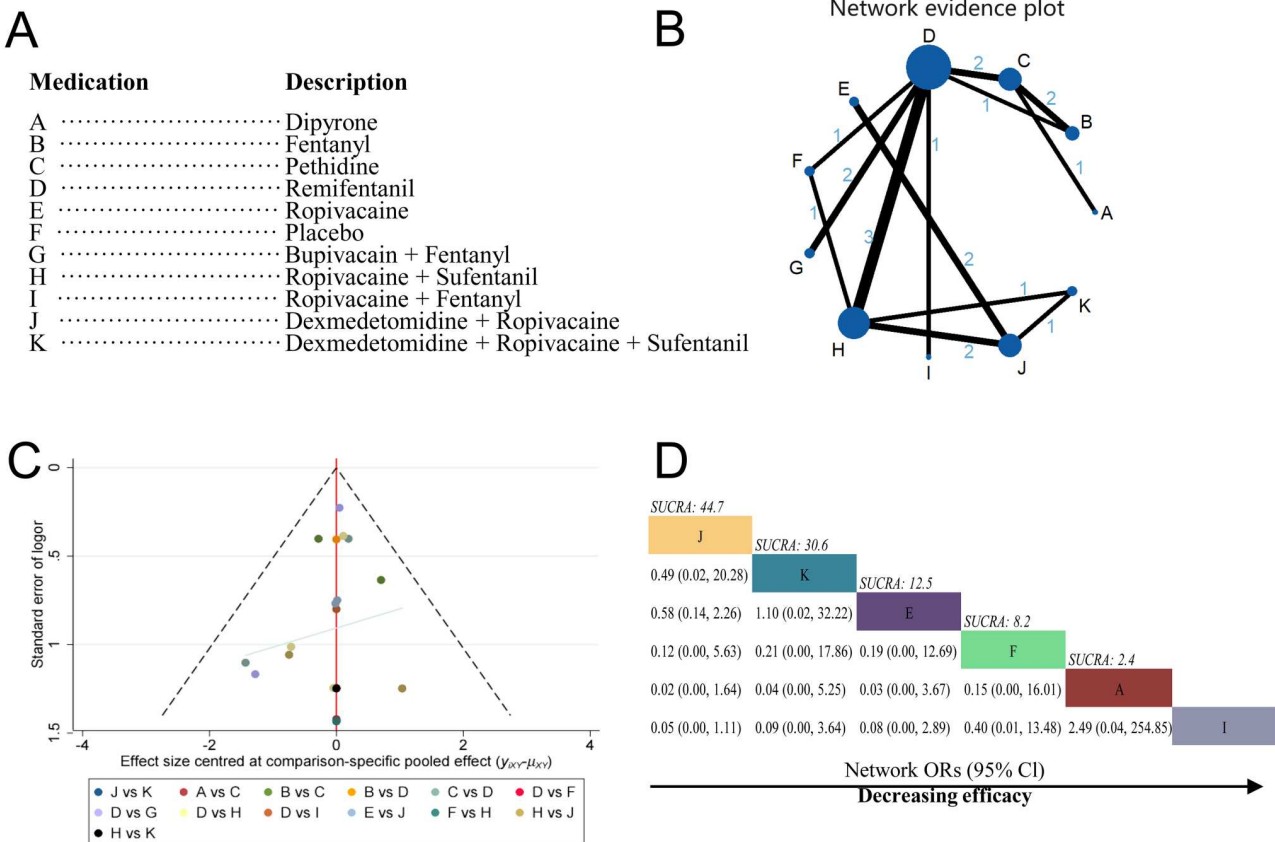

**Fig 5. Network meta-analysis by individual treatment.** (A) Description of drug categories included in the network meta-analysis. (B) Network graph showing overall treatment effect comparisons between nodes (blue circles), where each node represents a drug category or placebo. The size of each node is proportional to the total number of participants randomized to receive the drug category. The width of each connecting line is proportional to the number of trial-level comparisons between the two nodes. (C) Funnel plot of publication bias, comparing publication bias between drug categories. (D) Schematic diagram listing the most effective drug categories globally according to surface under the cumulative ranking curve (SUCRA) analysis. CI, confidence interval; OR, odds ratio.

In the network meta-analysis, the most interactions were between bupivacain+fentanyl and remifentanil (3 interactions), with the rest having either 1 or 2 interactions (Fig 5B). The $I^2$ value was 36%, indicating moderate heterogeneity (Fig 5C). Then a random-effects model was used for the network meta-analysis.

The results showed that compared to the combination of dexmedetomidine+ropivacaine, the use of bupivacaine+fentanyl resulted in a higher incidence of maternal nausea and vomiting reactions (Class G vs J; OR, 16.26; 95% CI, 1.15–334.03). Dexmedetomidine+ropivacaine demonstrated a lower incidence of nausea and vomiting reactions compared to fentanyl (Class J vs B; OR, 0.04; 95% CI, 0.00–0.54), sufentanil+ropivacaine (Class J vs H; OR, 0.13; 95% CI, 0.01–0.67), pethidine (Class J vs C; OR, 0.02; 95% CI, 0.00–0.23), and remifentanil (Class J vs D; OR, 0.04; 95% CI, 0.00–0.43). The combination of dexmedetomidine+ropivacaine+sufentanil showed a lower incidence of nausea and vomiting reactions compared to pethidine (Class K vs C; OR, 0.04; 95% CI, 0.00–0.88). pethidine had a higher incidence of nausea and vomiting reactions compared to ropivacaine (Class K vs C; OR, 31.54; 95% CI, 1.70–1040.36). There were no significant differences in the occurrence rates of adverse reactions between the remaining drug comparisons. Therefore, considering the incidence of maternal nausea and

vomiting adverse reactions, the combination of dexmedetomidine+ropivacaine is preferred, while pethidine has the most severe adverse reactions (Fig 5D). SUCRA analysis showed that dexmedetomidine+ropivacaine had the highest cumulative ranking for reducing the side effect of nausea and vomiting in parturients (Class J; SUCRA = 44.7%), followed by dexmedetomidine+ropivacaine+sufentanil (Class K; SUCRA = 30.6%), ropivacaine (Class E; SUCRA = 12.5%), placebo (Class F; SUCRA = 8.2%), and dipyrone (Class A; SUCRA = 2.4%; Fig 5D).

## Sensitivity analysis

To examine the reliability and robustness of the aforementioned results and conclusions, sensitivity analysis was performed by using alternative analysis models. The sensitivity analysis results showed that regardless of using a consistency model or an inconsistency model, the direction and significance of the conclusions remained unchanged (S2 Fig in S2 File).

## Discussion

This study represents a comprehensive systematic review and network meta-analysis of labor pain relief approaches for parturients. Fifteen high-quality RCTs evaluating 11 labor analgesics were included to assess effects on pain control and related outcomes. Compared to placebo, dexmedetomidine/ropivacaine/sufentanil and dexmedetomidine/ropivacaine combinations, bupivacaine/fentanyl and ropivacaine/sufentanil combinations, alongside remifentanil monotherapy, demonstrated superior pain alleviation and maternal comfort. Notably, dexmedetomidine/ropivacaine reduced adverse reactions such as nausea/vomiting versus alternatives. Collectively, intrathecal dexmedetomidine/ropivacaine administration effectively relieved labor pain while decreasing side effects, constituting an advisable approach for obstetric analgesia. Our findings impart crucial clinical and research references within this domain.

The focus on labor pain relief has led to several previous studies comparing the effects of analgesics during labor. Previous meta-analyses involving 8 studies by Myeongjong and colleagues and 5 studies by ZhiQiang and colleagues found that remifentanil PCA non-superior to epidural analgesia for labor pain relief [30, 31]. Specifically, the meta-analyses involving 12 studies by Schnabel and colleagues showed epidural analgesia conferred superior relief versus remifentanil [32], aligning with our conclusions. Previous examinations also demonstrated no remarkable difference in adverse events between remifentanil PCA and epidural anesthesia [30–32], consistent with our analysis. At present, few analyses compare multi-drug analgesic efficacies, with no consensus conclusions. This may owe to previous categorizations by anesthesia technique without drug specification. Indeed, numerous drug combinations exist for epidural analgesia. Our network meta-analysis specifically classified constituent drugs, firstly by opioid content, and secondly by individual agents. This permitted exclusion of inter-study dose variation influences. Our observation of lower dexmedetomidine epidural incidence of nausea/vomiting versus alternatives, similar to placebo levels, agrees with past meta-analyses by Nijuan and colleagues [29], validating our results.

Several advantages distinguish our work from preceding investigations. Notably, larger sample sizes from higher-quality RCTs improved result reliability. Our evaluation and comparison of an extensive range of analgesia techniques, including epidural, intramuscular and intravenous modalities, imparted comprehensive clinical guidance. Inclusion exclusively of RCTs with quantifiable pain criteria also enhanced accuracy. By concentrating on analgesic efficacy, our work promotes enhanced labor experiences for parturients. Assessment of nausea/vomiting supplements a comprehensive profile of risk-benefit considerations. Nevertheless, certain limitations persist, including omission of key neonatal outcomes, inter-study

variations in scales/timepoints that may influence accuracy, numerical translation of pain scores, geographical constraints from the preponderance of Chinese trials, heterogeneity introduced via intervention/dose differences, and subjectivity in processes such as screening or bias assessment. Publication bias may also confound results.

## Conclusion

The combination of opioids and non-opioids or the combination of dexmedetomidine, ropivacaine, and sufentanil effectively relieved labor pain. Dexmedetomidine combined with ropivacaine has a lower incidence of adverse reactions of nausea and vomiting than other methods. However, considering the limitations of the research, the conclusions should still be interpreted with caution. Further high-quality randomized controlled trials are needed, particularly focusing on the evaluation of neonatal outcomes.

## Supporting information

**S1 File. Search strategy.**
(PDF)

**S2 File. Supplementary figures and tables.**
(PDF)

**S3 File. PROSPERO and other supplementary information.**
(PDF)

**S1 Checklist. PRISMA NMA checklist of Items to include when reporting a systematic review involving a network meta-analysis.**
(DOCX)

## Acknowledgments

The authors thank the experts who responded kindly to specific questions during the conduct of this research.

## Author Contributions

**Conceptualization:** Yiru Chen, Hongchun Chen.

**Data curation:** Yiru Chen, Chunhui Yuan.

**Formal analysis:** Yiru Chen, Chunhui Yuan.

**Investigation:** Yiru Chen.

**Methodology:** Yiru Chen.

**Project administration:** Chunhui Yuan.

**Software:** Yiru Chen, Hongchun Chen, Chunhui Yuan.

**Supervision:** Hongchun Chen, Chunhui Yuan.

**Validation:** Hongchun Chen.

**Visualization:** Hongchun Chen.

**Writing – original draft:** Yiru Chen.

**Writing – review & editing:** Yiru Chen, Hongchun Chen, Chunhui Yuan.

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
