## [Decision Letter · Decision Letter 0]

4 Mar 2024

PONE-D-24-00615Comparative Efficacy of Opioid and Non-Opioid Analgesics in Labor Pain Management: A Network Meta-AnalysisPLOS ONE

Dear Dr. Yuan,

Thank you for submitting your manuscript to PLOS ONE. After careful consideration, we feel that it has merit but does not fully meet PLOS ONE’s publication criteria as it currently stands. Therefore, we invite you to submit a revised version of the manuscript that addresses the points raised during the review process.

We look forward to receiving your revised manuscript.

Kind regards,

Stefano Turi

Academic Editor

PLOS ONE

Journal Requirements:

Reviewers' comments:

Reviewer's Responses to Questions

**Comments to the Author**

1. Is the manuscript technically sound, and do the data support the conclusions?

Reviewer #1: Yes

Reviewer #2: Yes

2. Has the statistical analysis been performed appropriately and rigorously? 

Reviewer #1: I Don't Know

Reviewer #2: Yes

3. Have the authors made all data underlying the findings in their manuscript fully available?

Reviewer #1: Yes

Reviewer #2: Yes

4. Is the manuscript presented in an intelligible fashion and written in standard English?

Reviewer #1: Yes

Reviewer #2: Yes

5. Review Comments to the Author

Reviewer #1: Consider making the title more specific and concise. For example, "Comparative Efficacy and Safety of Analgesics for Labor Pain Management: A Meta-Analysis."

The background section is well-written. However, consider adding a brief statement about the importance of effective labor pain management for both the mother and the baby.

Clearly state the primary and secondary objectives of the meta-analysis at the end of the introduction.

Consider including a flowchart to visually represent the study selection process, similar to the PRISMA flow diagram.

Clearly list the inclusion and exclusion criteria in a concise manner. Explain the rationale behind each criterion.

Clearly describe the statistical methods used, including how heterogeneity was assessed and managed. Clarify how heterogeneity was assessed, and consider providing the statistic for each analysis.

For the network meta-analysis, consider using node-splitting analysis or other methods to evaluate the consistency of the network. This is crucial for assessing the reliability of indirect treatment comparisons.

In the safety analysis, if applicable, consider using absolute risk differences along with odds ratios to convey a more comprehensive understanding of safety outcomes.

Provide a summary of the risk-of-bias assessment results. Consider including a table summarizing the bias assessment for each included study.

Clearly define the primary and secondary outcome measures in results

Present the network meta-analysis results in a clear and organized manner. Consider using forest plots to visually represent the effect sizes and confidence intervals in results

Discuss the clinical implications of the findings. Compare your results with previous studies and discuss any inconsistencies.Clearly outline the limitations of the meta-analysis.

In conclusion: Clearly restate the key findings and their implications for labor pain management. Consider avoiding the term "Notably" in your conclusion unless you can explicitly highlight a unique or unexpected finding

Reviewer #2: Page & Line No. Comments

Abstract

Background

Page 2 & Line no 1-4 it is advisable to give strong insights of the categorization/ classification of drugs in the background to increase readability

Methodology

Page 2 & Line No 5 Kindly mention the restricted time frame in the search strategy (Starting year to ending year)

“Fifteen randomized controlled trials evaluating analgesic interventions in ASA I or II parturients were included” Please incorporate these findings into the results section.

Introduction

Page 3 & Line no 1-4 Please describe the class of medications used to treat labor pain, their mode of action, and how they are administered.

Page no 3 & Line no 5-6 Kindly enlist the American Society of Anaesthesiologists (ASA) physical status classification system levels

Methods

Page 4 & line no 16-18 Kindly mention the PROSPERO registration number.

“Assessing the methodological quality of systematic reviews (AMSTAR-2)(13) Guidelines” Kindly re-check the sentence AMSTAR-2 was not intended to deal with Meta-analysis of individual patient data or Network Meta-analysis

Page 5 & Line no 3-7 Kindly mention the number of results in the detailed search strategy for every search (1 Population & 2 Intervention) in supplementary files

Page 7 & Line no 2-6 “This tool evaluates five bias domains: random sequence generation,

allocation concealment, participant/personnel blinding, incomplete outcome data, and selective outcome reporting”. Kindly write the domains as per the ROB 2 tool

Kindly specify pre-defined criteria for some concern/some degree of risk

Page no 7 & 13-14 Kindly specify the type of package used in R Software for Network meta-analysis

Results

Page 9 & Line no 1-2 Kindly check the PRISMA flow chart for the total excluded studies (597) in the primary screening. (267 not relevant, 148 duplicates 82 reviews, 37 conferences, theses, books, posters, and 27 case reports, the sum of these primary excluded studies (561) is not matched to the total excluded studies)

Page 10 Kindly provide legends to the treatment groups in study characters

Page 14 & Line No 1-5 Kindly check the percentage proportion of some concerns out of 15 studies.

Kindly provide a summary (Overall studies) of ROB apart from individual studies.

Kindly provide reasons/explanations for some concerns (Some degree of bias risk) in respective studies

Page 14 & Line No 15 Kindly check the number of interactions in the network plot between combination therapy and placebo

It is advisable to use the word Network Meta-Analysis instead of Meta-Analysis in the entire manuscript

kindly avoid the sentence “please insert the figure here” in the entire manuscript

The manuscript is poorly written. Please revise the manuscript in light of the comments. Overall, this manuscript requires major modification and can only be continued after extensive revision.

6. PLOS authors have the option to publish the peer review history of their article (what does this mean?). If published, this will include your full peer review and any attached files.

Reviewer #1: No

Reviewer #2: No

---

## [Author Response · Author response to Decision Letter 0]

17 Mar 2024

Dear Dr. Stefano Turi and reviewers:

On behalf of my co-authors, we appreciate editor and reviewers very much for their constructive comments and suggestions on our manuscript [PONE-D-24-00615] entitled “Comparative Efficacy of Opioid and Non-Opioid Analgesics in Labor Pain Management: A Network Meta-Analysis”. We promise to provide our data at the time of acceptance and not to modify our previous statements.

We have studied the comments carefully and made corrections which we hope meet with approval. All the changes in the manuscript are marked in red. The corrections are in the manuscript and the responds to the reviewers’ comments are as follows.

Replies to reviewer’s comments：

Reviewer #1:

Comment 1: Consider making the title more specific and concise. For example, "Comparative Efficacy and Safety of Analgesics for Labor Pain Management: A Meta-Analysis."

Respond: Thank you for your suggestion. After careful consideration, we think that the current title may highlight the grouping expression of opioid and non-opioid drugs, so we did not adapt the relatively concise title.

Comment 2: The background section is well-written. However, consider adding a brief statement about the importance of effective labor pain management for both the mother and the baby.

Respond: Thank you for your compliment. We have added a brief statement in the background to make the manuscript more readable.

Comment 3: Clearly state the primary and secondary objectives of the meta-analysis at the end of the introduction.

Respond: Thank you for your suggestion. We have added a statement about the primary and secondary objectives of the meta-analysis at the end of the introduction.

Comment 4: Consider including a flowchart to visually represent the study selection process, similar to the PRISMA flow diagram.

Respond: Again thank you for your advice. We made a more regulated PRISMA flow diagram named Fig 1.

Comment 5: Clearly list the inclusion and exclusion criteria in a concise manner. Explain the rationale behind each criterion.

Respond: Thank you for your suggestion. The inclusion and exclusion criteria are set out on page six of the manuscript. Inclusion criteria are analyzed from four aspects: 1. Study population2. Study design3. Intervention4. Outcome measures.

Comment 6: Clearly describe the statistical methods used, including how heterogeneity was assessed and managed. Clarify how heterogeneity was assessed, and consider providing the statistic for each analysis.

Respond: Thank you for your suggestion. Statistical analysis was performed on page seven of the manuscript, Include Quantitative heterogeneity across trials was assessed using the I2 statistic. In case of low heterogeneity (I2<50%), a fixed-effects model was utilized for meta-analysis. Otherwise, a random-effects model was employed for sensitivity analysis and subgroup analysis to explore potential sources of heterogeneity ". We appreciate the reviewer's careful evaluation. Thank you again for the careful review of our manuscript.

Comment 7: For the network meta-analysis, consider using node-splitting analysis or other methods to evaluate the consistency of the network. This is crucial for assessing the reliability of indirect treatment comparisons.

Respond: Thank you for the opportunity to clarify our approach to evaluating inconsistency. Since the network meta-analysis contained no loops, node-splitting analysis could not be conducted. However, to thoroughly assess inconsistency, inconsistency models were fitted and inconsistency factor estimates with 95% CIs were reported in the Supporting information S2. No significant differences from the consistency model were observed upon inspection. While node-splitting was not feasible given the loop-free structure, comparing consistency and inconsistency model fits can still provide valuable information about potential inconsistency. We appreciate the reviewer's careful evaluation. Thank you again for the careful review of our manuscript.

Comment 8: In the safety analysis, if applicable, consider using absolute risk differences along with odds ratios to convey a more comprehensive understanding of safety outcomes.

Respond: Thank you for your review of our manuscript and for providing valuable suggestions. We appreciate the recommendation to utilize absolute risk differences and odds ratios for conveying a more comprehensive understanding of safety outcomes. However, we believe that adopting these measures within our current analytical framework may add complexity rather than simplify understanding. We have thoroughly considered various approaches and have chosen the method that we believe will most clearly communicate the safety results. Once again, we appreciate your valuable input.

Comment 9: Provide a summary of the risk-of-bias assessment results. Consider including a table summarizing the bias assessment for each included study.

Respond: Thank you for your advice. We provide an assessment of bias for each study in Supporting information S2 Fig 1.

Comment 10: Clearly define the primary and secondary outcome measures in results

Respond: Thank you for your advice. In The Study selection of the manuscript result, the definition of primary and secondary results was added. "The primary outcome measure was the pain relief score calculated from the pain ratings (using the visual analog scale, VAS[14], or numeric rating scale, NRS[1]) at 20, 30, or 60 minutes after the onset of the first stage of labor or at the start of the second stage of labor. The secondary outcome of interest is drug side effects, such as nausea and vomiting." 

Comment 11: Present the network meta-analysis results in a clear and organized manner. Consider using forest plots to visually represent the effect sizes and confidence intervals in results

Respond: Thank you for your advice. We generated forest plots and put it in Supporting information S2, named S2 Fig2 and S2 Fig3.

Comment 12: Discuss the clinical implications of the findings. Compare your results with previous studies and discuss any inconsistencies. Clearly outline the limitations of the meta-analysis.

Respond: Thank you for your suggestion. In the discussion section, we compared our results with previous meta-analyses, and listed the possible limitations of this network meta-analysis in detail.

Comment 13: In conclusion: Clearly restate the key findings and their implications for labor pain management. Consider avoiding the term "Notably" in your conclusion unless you can explicitly highlight a unique or unexpected finding.

Respond: Thank you for your suggestion. The conclusion was elaborated upon, with the removal of the term "notably" to enhance precision in expression.

Reviewer #2: 

Comment 1:Page 2 & Line no 1-4 it is advisable to give strong insights of the categorization/ classification of drugs in the background to increase readability

Respond: Thank you for your suggestion. The note "classified by drug category and individual treatment methods" was added to increase readability.

Comment 2: Page 2 & Line No 5 Kindly mention the restricted time frame in the search strategy (Starting year to ending year)

Respond: Thank you for your suggestion. Added starting time to manuscript "All searches commenced from the database's inception to the date of the literature search (May 31, 2023)."

Comment 3: “Fifteen randomized controlled trials evaluating analgesic interventions in ASA I or II parturients were included” Please incorporate these findings into the results section.

Respond: Thank you for your suggestion. These findings have been incorporated into the results section.

Comment 4: Page 3 & Line no 1-4 Please describe the class of medications used to treat labor pain, their mode of action, and how they are administered.

Respond: Thank you for your suggestion. A description of the class of medications used to treat labor pain, their mode of action, and how they are administered has been added to the third page of the manuscript

Comment 5: Page no 3 & Line no 5-6 Kindly enlist the American Society of Anaesthesiologists (ASA) physical status classification system levels

Respond: Thank you for your suggestion. A detailed definition of the physical status classification system level of the American Society of Anesthesiologists (ASA) has been added on page 4 of the manuscript

Comment 6: Page 4 & line no 16-18 Kindly mention the PROSPERO registration number.

Respond: Thank you for your suggestion. The PROSPERO registration number "CRD42023417670" has been added.

Comment 7: “Assessing the methodological quality of systematic reviews (AMSTAR-2)(13) Guidelines” Kindly re-check the sentence AMSTAR-2 was not intended to deal with Meta-analysis of individual patient data or Network Meta-analysis

Respond: Thank you for pointing out this error. We have deleted the AMSTAR-2 related methods mentioned in the manuscript. Thank you again for your careful review of the manuscript.

Comment 8: Page 5 & Line no 3-7 Kindly mention the number of results in the detailed search strategy for every search (1 Population & 2 Intervention) in supplementary files

Respond: Thank you for your suggestion. The number of results in the detailed search strategy for every search (1 Population & 2 Intervention) in Supporting information has been mentioned in the Supporting information S1.

Comment 9: Page 7 & Line no 2-6 “This tool evaluates five bias domains: random sequence generation, allocation concealment, participant/personnel blinding, incomplete outcome data, and selective outcome reporting”. Kindly write the domains as per the ROB 2 tool

Respond: Thank you for your suggestion. We have written the domains as per the ROB2 tool and added the detailed content in the Supplementary ROB2_Summary.

Comment 10: Kindly specify pre-defined criteria for some concern/some degree of risk

Respond: Thank you for your suggestion. The ROB 2 tool definition standard has been added at the end of this section. "Judgments are based on, and summarise, the answers to signalling questions."

Comment 11: Page no 7 & 13-14 Kindly specify the type of package used in R Software for Network meta-analysis

Respond: Thank you for your suggestion. Package "gemtc" and "netmeta" used in R have been supplemented.

Comment 12: Page 9 & Line no 1-2 Kindly check the PRISMA flow chart for the total excluded studies (597) in the primary screening. (267 not relevant, 148 duplicates 82 reviews, 37 conferences, theses, books, posters, and 27 case reports, the sum of these primary excluded studies (561) is not matched to the total excluded studies)

Respond: Thank you for pointing out the mistake. It has been remade according to the form of PRISMA 2020 flow diagram to make it look more concise and clear.

Comment 13: Page 10 Kindly provide legends to the treatment groups in study characters

Respond: Thank you for your suggestion. We have carefully considered your suggestions. In our opinion, The characteristics of included studies can be better described in the form of tables. We have made "Table 1:The characteristics of included studies."

Comment 14: Page 14 & Line No 1-5 Kindly check the percentage proportion of some concerns out of 15 studies.

Respond: Thank you for pointing out the mistake. It has been changed from 53% to 47%.

Comment 15: Kindly provide a summary (Overall studies) of ROB apart from individual studies.

Respond: Thank you for your advice. A summary of ROB was provided in the Supplementary ROB2_Summary..

Comment 16: Kindly provide reasons/explanations for some concerns (Some degree of bias risk) in respective studies

Respond: Thank you for your advice. The Explanations for some concerns were provided in the Supplementary ROB2_Summary.

Comment 17: Page 14 & Line No 15 Kindly check the number of interactions in the network plot between combination therapy and placebo

Respond: Thank you for pointing out the error, we checked and corrected the number of interactions in the network plot between combination therapy and placebo,

Comment 18: It is advisable to use the word Network Meta-Analysis instead of Meta-Analysis in the entire manuscript

Respond: Thank you for your suggestion. The term "meta-analysis of networks" has been used instead of "meta-analysis" in the entire manuscript.

Comment 19: Kindly avoid the sentence “please insert the figure here” in the entire manuscript

Respond: Thank you for your suggestion. The sentence "please insert the figure here" used in the manuscript was removed.

We would like to express our great appreciation to you and reviewers for comments on our paper. Looking forward to hearing from you.

Thank you and best regards.

Yours sincerely, 

Prof. Chunhui Yuan 

Address: Key Laboratory of Novel Targets and Drug Study for Neural Repair of Zhejiang Province, School of Medicine, Hangzhou City University, 310015, Hangzhou, People’s Republic of China. Tel.: +86 0571 88281303.

E-mail: ch_yuan@zju.edu.cn(CH.Y)

---

## [Decision Letter · Decision Letter 1]

22 Apr 2024

Comparative Efficacy of Opioid and Non-Opioid Analgesics in Labor Pain Management: A Network Meta-Analysis

PONE-D-24-00615R1

Dear Dr. Chunhui Yuan,

We’re pleased to inform you that your manuscript has been judged scientifically suitable for publication and will be formally accepted for publication once it meets all outstanding technical requirements.

Kind regards,

Stefano Turi

Academic Editor

PLOS ONE

I did not receive comments by reviewer 2, despite several invitations. However, in my opinion the authors addressed all the comments proposed by reviewer 2.

Below the comments by reviewer 1. 

Reviewers' comments:

Reviewer's Responses to Questions

**Comments to the Author**

1. If the authors have adequately addressed your comments raised in a previous round of review and you feel that this manuscript is now acceptable for publication, you may indicate that here to bypass the “Comments to the Author” section, enter your conflict of interest statement in the “Confidential to Editor” section, and submit your "Accept" recommendation.

Reviewer #1: All comments have been addressed

2. Is the manuscript technically sound, and do the data support the conclusions?

Reviewer #1: Yes

3. Has the statistical analysis been performed appropriately and rigorously? 

Reviewer #1: Yes

4. Have the authors made all data underlying the findings in their manuscript fully available?

Reviewer #1: Yes

5. Is the manuscript presented in an intelligible fashion and written in standard English?

Reviewer #1: Yes

6. Review Comments to the Author

Reviewer #1: All the chnages that had been advised are fully incorporated in revised version. This is now comnplete and good manuscript to publish.

7. PLOS authors have the option to publish the peer review history of their article (what does this mean?). If published, this will include your full peer review and any attached files.

Reviewer #1: **Yes: **Dr. LALIT GUPTA

---

## [Editor Report · Acceptance letter]

24 May 2024

PONE-D-24-00615R1 

PLOS ONE

Dear Dr. Yuan, 

I'm pleased to inform you that your manuscript has been deemed suitable for publication in PLOS ONE. Congratulations! Your manuscript is now being handed over to our production team.

Kind regards, 

on behalf of

Dr. Stefano Turi 

Academic Editor

PLOS ONE